# When RL Meets Adaptive Speculative Training:
# A Unified Training-Serving System

**Junxiong Wang** [* † 1] **Fengxiang Bie** [* † 2] **Jisen Li** [† 1] **Zelei Shao** [† 3] **Qingyang Wu** [1] **Yinghui Liu** [1] **Yubo Wang** [1]
**Avner May** [1] **Ben Athiwaratkun** [1] **Yineng Zhang** [1] **Shuaiwen Leon Song** [2] **Zhongzhu Zhou** [† 1 2] **Chenfeng Xu** [† 4 1]
**Xiaoxia Wu** [† 1]

## Abstract

Speculative decoding can significantly accelerate LLM serving, yet most deployments today disentangle speculator training from serving, treating speculator training as a standalone offline modeling problem. We show that this decoupled formulation introduces substantial deployment and adaptation lag: (1) *high time-to-serve*, since a speculator must be trained offline for a considerable period before deployment; (2) *delayed utility feedback*, since the true end-to-end decoding speedup is only known after training and cannot be inferred reliably from acceptance rate alone due to model-architecture, diverse prompt engineering, and system-level overheads; and (3) *domain-drift degradation*, as the target model is repurposed to new domains and the speculator becomes stale and less effective. To address these issues, we present *Aurora*, a unified training–serving system that closes the loop by continuously learning a speculator directly from live inference traces. *Aurora* reframes online speculator learning as an asynchronous reinforcement-learning problem: accepted tokens provide positive feedback, while rejected speculator proposals provide implicit negative feedback that inherits the online traffic failure signals. Our design integrates an SGLang-based inference server with an asynchronous training server, enabling hot-swapped speculator updates without service interruption. Crucially, *Aurora* supports day-0 deployment: a speculator can be served immediately and rapidly adapted to live traffic, improving system

performance while providing immediate utility feedback. Across experiments, *Aurora* achieves a $1.5\times$ day-0 speedup on recently released frontier models (e.g., MiniMax M2.1 229B and Qwen3-Coder-Next 80B). *Aurora* also adapts effectively to distribution shifts in user traffic, delivering an additional $1.25\times$ speedup over a well-trained but static speculator on widely used models (e.g., Qwen3 and Llama3).

## 1. Introduction

The deployment of large language models (LLM) is increasingly strained by inference costs (Google Cloud Blog, 2025; OpenAI, 2026; Anthropic, 2026). Speculative decoding (Leviathan et al., 2023; Li et al., 2024b; 2025; Chen et al., 2023; Fu et al., 2024) is a compelling system lever: a lightweight drafter proposes multiple tokens and a high-quality verifier (the target model) checks and accepts a prefix. This reduces the number of expensive target-model decode steps while preserving output quality (Miao et al., 2023; Cai et al., 2023; Chen et al., 2024). When the drafter's proposals align closely with the verifier, speculative decoding can deliver substantial speedups at the production scale (Xia et al., 2024; Zhang et al., 2024).

Most speculative decoding deployments today follow a two-stage pipeline: (i) an *offline training workflow* that trains a drafter to maximize token acceptance via supervised targets and/or off-policy distillation from target-model activations (Li et al., 2024b; 2025), and (ii) a separate *serving stack* that loads the trained drafter and performs speculative decoding online. While this separation is organizationally convenient, we argue that under real-world deployment constraints, speculative decoding must be reconsidered jointly from both algorithmic and system perspectives.

### 1.1. Algorithm Gap: Speculative Training Should Pivot Toward Modeling Local Traffic

Prior work implicitly formulates speculative training as large-scale model distillation: the drafter is optimized to

---

[*]Equal Contribution. [†]Core Contributors. The project leads are Xiaoxia Wu, Chenfeng Xu, and Junxiong Wang. [1]Together AI [2]University of Sydney, Australia [3]University of Illinois Urbana-Champaign, USA [4]University of Texas at Austin, USA. Correspondence to: Xiaoxia Wu <shirley@together.ai>.

*Proceedings of the 43rd International Conference on Machine Learning*, Seoul, South Korea. PMLR 306, 2026. Copyright 2026 by the author(s).

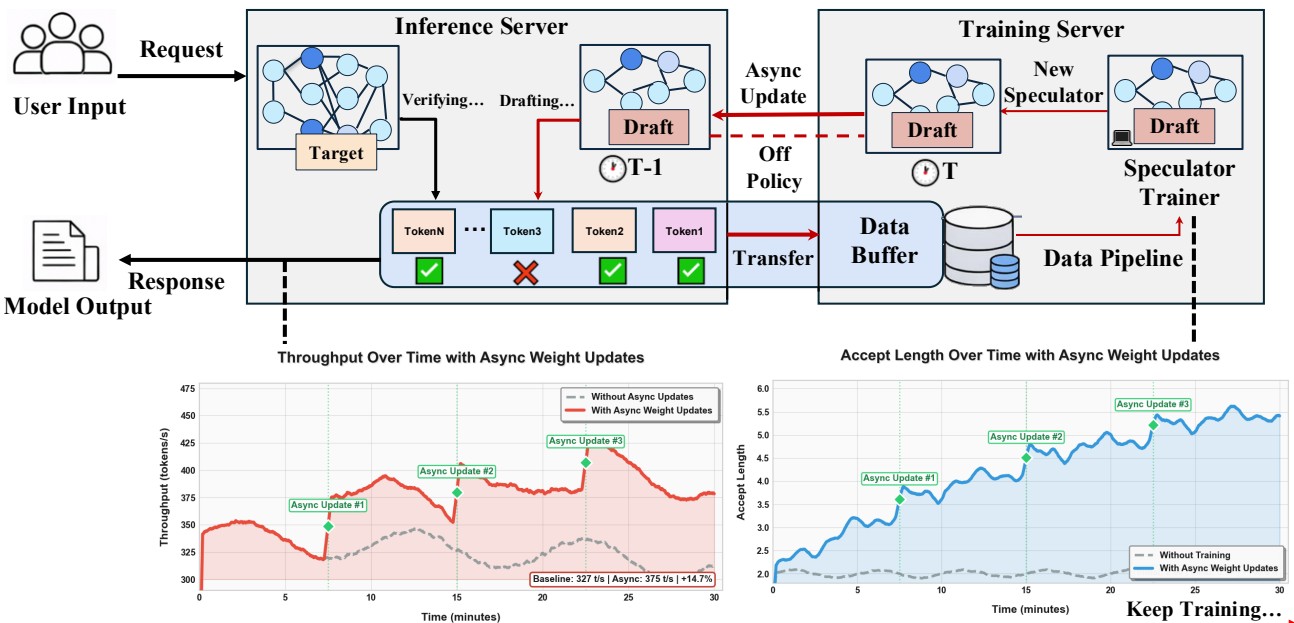

*Figure 1.* **Aurora.** A unified training–serving framework for online speculative training with asynchronous, RL-style updates. A production inference server performs speculative decoding with a fixed target (verifier) and a lightweight draft model (speculator), accepting or rejecting proposed tokens during verification. Serving traces—including both accepted and rejected prefixes—are streamed into a data buffer and training pipeline. A separate training server continuously updates the speculator from collected off-policy data and periodically hot-swaps asynchronous model updates into the inference server without interrupting requests. Bottom: (left) per-request throughput over time, exhibiting step changes after each asynchronous update; (right) acceptance-length statistics during continuous training, showing improving (or sustained) acceptance over time.

approximate the target model over a large, static, and global offline data distribution. In this view, speculative decoding becomes a miniature replica of large-model pretraining: the target model fits massive datasets, while the drafter fits the target's behavioral traces (e.g., logits or activations) across the same broad distribution.

However, this algorithm abstraction mismatches real-world serving. In deployment, inference traffic is neither global nor stationary, (1) production systems operate under *local time-varying traffic distributions*, and performance is determined by the *current request stream*, not the full behavioral manifold of the target model. A drafter that is globally optimal with respect to the target model's distribution may be suboptimal or even inefficient under the specific workload encountered at serving time. (2) Even worse, in practical deployments, the target model itself is not static, it may frequently be updated for quality, safety, cost optimization, or hardware migration. In contrast, the drafter is typically refreshed on a much slower cadence due to retraining cost and pipeline complexity. This creates distributional staleness: the deployed drafter tracks an outdated verifier distribution, leading to degraded acceptance rates and diminishing speculative gains over time. Motivated by this gap, we propose to fundamentally redefine speculative training for real-world settings:

**Instead of pursuing global optimality over the target model's full behavioral space, we shift the objective toward rapid adaptation to a locally optimal drafter under the current traffic distribution.**

Under this formulation, speculative training becomes a *dynamic systems problem*, in which learning objectives are tightly coupled with serving-time workload characteristics and deployment constraints.

### 1.2. System Gap: Speculative Training Should Be Treated as a Dynamic System

Despite this need for adaptivity, existing speculative training pipelines remain fundamentally static. This mismatch also introduces several system-level frictions in real-world deployments:

**(1) Day-0 support is difficult.** New frontier models are released on a weekly cadence (LLM Stats, 2026), and operators expect inference acceleration from day 0 of deployment. However, static speculative systems typically require lengthy calibration and retraining cycles for the drafter before delivering reliable gains. This introduces substantial wall-clock delay and engineering overhead, increasing time-to-serve and limiting operational agility.

**(2) Training–serving mismatch delays utility feedback.** Offline drafter training may maximize token acceptance in isolation, yet production speedups are ultimately determined by the entire deployment stack, including kernel implementations, numeric precision (e.g., FP8/FP4), batching and scheduling policies, prompt diversity, and even microarchitectural effects. In practice, the drafter and verifier frequently operate under different runtime constraints and may even originate from different model families. Such mismatches introduce incompatibilities, distribution drift, and unexpected throughput regressions. As a result, true end-to-end speedup cannot be reliably predicted without serving-time evaluation.

**(3) High infrastructure cost.** Drafter calibration pipelines often require collecting large volumes of target-model activations or related signals by executing the verifier over a distillation corpus. These intermediate artifacts must be stored, versioned, transferred (often across regions or clusters), and replayed during training. At production scale, the storage footprint can reach petabyte-level magnitude, incurring significant memory, bandwidth, and operational complexity costs.

### 1.3. Aurora: A Unified Training-Serving Speculative System

These joint algorithm-system gaps suggest that speculative decoding should not be treated merely as a modeling problem ("train a better drafter"), but as a **joint learning-and-serving problem**. Inspired by modern reinforcement learning (RL) systems that tightly integrate online inference with continuous training, we propose an **asynchronous unified training-service system with speculative decoding**, called ***Aurora*** (Figure 1) that co-designs drafter learning and inference within a single integrated system.[1]

A key observation is that speculative decoding admits a natural asynchronous RL-style framing. The drafter acts as a policy that proposes token sequences; the verifier returns structured feedback through accept/reject decisions; and the objective is to maximize expected acceptance (equivalently, maximize verified tokens per verifier step) under the deployed distribution. Concretely, we place a trainable speculator on dedicated training resources while deploying a shadow copy of the speculator on the serving node. As live traffic arrives, the serving node continuously collects model inputs, accepted/rejected draft outcomes, and lightweight supervision signals and writes them into a bounded memory buffer. Once sufficient data accumulates, the training node updates the speculator on these freshly collected trajectories and periodically ships a new speculator snapshot back to the

serving node. This design enables online adaptation of the speculator under real traffic while keeping the service stable and predictable.

Although the architecture resembles asynchronous RL, speculative training operates under fundamentally different objectives and constraints:

1. **The objective is serving efficiency, not rollout throughput.** We optimize end-to-end latency, tokens-per-second, and cost per output token under strict SLO constraints, rather than rollout throughput for policy improvement.
2. **Lazy, non-disruptive synchronization.** Frequent weight pushes can disrupt service (cache invalidation, latency jitter, transient regressions), so we schedule updates lazily to keep inference stable rather than synchronizing aggressively.
3. **Domain shift is the primary failure mode.** Unlike standard RL rollouts with a consistent interaction loop, speculative decoding quality can degrade sharply under even mild drafter–verifier mismatch between offline training and live traffic.

**Contribution.** *Aurora* represents a paradigm shift for speculative decoding: rather than treating training as an offline prerequisite and serving as a fixed endpoint, we close the loop and make the speculator continuously improvable under live traffic. This unified training–serving design unlocks capabilities difficult to achieve in the standard pipeline:

1. **Day-0 serving and real-time observability.** A speculator can be deployed from scratch and adapted *in situ* from on-policy serving data, reframing speculative decoding from a "train-then-serve" workflow into a *serve-to-train* flywheel.
2. **Fast adaptation that mitigates distribution mismatch.** Online data collection and a domain-controlled buffer reduce training–serving shift, yielding a $1.25\times$ improvement over a strong offline baseline and indicating that mismatch is the dominant failure mode.
3. **Lower infrastructure footprint.** Training under live traffic eliminates the large-scale activation-collection and replay pipelines used to distill from the target model.
4. **Scalable and algorithm-agnostic.** *Aurora* supports heterogeneous request mixtures (online traffic plus offline corpora), scales to large GPU deployments via RL-inspired distributed patterns, and interfaces cleanly with a broad family of speculative-decoding variants.

This paper is organized as follows: we discuss the preliminaries of speculative decoding and prior work on online speculative decoding in Section 2. We present our unified single-system design in Section 3. We then introduce our

---

[1]Code is available at https://github.com/togethercomputer/aurora.

framework for training speculators from scratch in Section 4. We also study how to adapt an existing speculator under domain shift in Section 4. We perform ablation studies under different serving configurations in Section 5. Finally, we scale up our approach with the most recent open-source frontier models in Section 6.

## 2. Background and Preliminaries

### 2.1. Speculative Decoding

Speculative decoding (Leviathan et al., 2023) is a fast-inference technique that pairs a lightweight draft model with a stronger target model. The draft proposes multiple tokens ahead and the target verifies them, accepting the longest matching prefix, so that the system can skip many expensive target-model decode steps while preserving output quality. In practice, the speedup depends on how closely the draft's token proposals align with the target (i.e., the acceptance rate). Build on top of this, MTP and Eagle (Li et al., 2024a;b; 2025) are the most popular frameworks, since they utilize the hidden state from target model which can boost acceptance rate. A key property of speculative decoding is that its expected speedup is largely determined by the average acceptance rate $\alpha$ and the lookahead $\gamma$ (the number of draft tokens proposed per iteration). Under the i.i.d. acceptance approximation, the expected number of tokens produced per verifier step is

$$\mathbb{E}[L] = \sum_{i=0}^{\gamma} \alpha^i = \frac{1 - \alpha^{\gamma+1}}{1 - \alpha}. \tag{1}$$

If we also account for draft overhead via the cost ratio $c = T_q/T_p$ (draft-step time over target-step time), the expected wall-clock improvement factor is approximately

$$\text{Speedup} \approx \frac{\mathbb{E}[L]}{1 + \gamma c} = \frac{1 - \alpha^{\gamma+1}}{(1 - \alpha)(1 + \gamma c)}. \tag{2}$$

In practice, however, accurately estimating $c$ is non-trivial: it depends on the serving stack (kernels, precision, batching/scheduling, and hardware) and can vary substantially under training–serving mismatch. Deeper draft models can achieve higher acceptance rates but run more slowly, while shallower models are faster but usually have lower acceptance rates. In practice, it's hard to weigh these trade-offs and choose a model. As a result, most teams train multiple drafters, but end up selecting only one. In contrast, our system enables a direct speedup comparison because it operates online.

### 2.2. Online Speculative Decoding

Online Speculative Decoding (OSD) (Liu et al., 2023) proposes a compelling variant of speculative decoding. We are inspired by this work and highlight the key differences.

From a utility and deployment standpoint, *Aurora* is a *closed-loop* unified training–inference system that operationalizes speculative decoding as a continuously improving service. Concretely, it closes three feedback paths: (1) the speculator trains online on streaming verifier outputs; (2) newly trained speculators are pushed back into the serving stack; and (3) real-time serving signals are fed back into training to concentrate learning where it yields measurable benefit. In contrast, OSD is effectively *open-loop*: it addresses (1) via online distillation from the target model, but does not close the loop through (2) and (3). Bridging these missing links is non-trivial, requiring co-design across both systems and algorithms.

1. On the **system** side, closing the loop means making model refresh efficient and robust under strict latency SLOs. This requires **(a)** a synchronization policy that rolls draft models into inference server without disrupting inference; **(b)** low-overhead mechanisms for transporting weights and training data so that refresh and replay do not become bottlenecks; and **(c)** a training pipeline that is efficient in streaming on-policy data, controls staleness, and remains stable as the request distribution changes.

2. On the **algorithmic** side, the key question is which training signal matches the production utility. OSD uses sequence-level distillation from target activations, but that signal can be poorly correlated with end-to-end speedup once architectural mismatch and system overhead dominate. Instead, we formulate training as *asynchronous RL*: accept/reject outcomes provide rewards that directly reflect real-time traffic and the verifier's online behavior. This aligns optimization with the online acceptance dynamics and the resulting throughput/latency, making the closed loop both practical and effective in deployment.

### 2.3. Asynchronous Reinforcement Learning and Online Training Systems

Asynchronous RL has emerged as a practical system paradigm for scaling RL post-training, motivated by the observation that rollout generation is often the dominant bottleneck in long and heavy-tailed trajectories, where synchronous pipelines suffer from stragglers and GPU idle time (Shao et al., 2025; Wang et al., 2025). By decoupling inference/rollout generation from training/optimization, asynchronous RL removes global synchronization barriers and improves end-to-end training efficiency, typically via a serverized actor setup that continuously produces experience while learners update in the background. AReaL (Fu et al., 2025) exemplifies this paradigm by providing an open-source framework that explicitly splits training and inference into separate components, together with algo-

rithmic designs that preserve training performance under asynchronization. In parallel, open-source systems such as slime (THUDM, 2024) and miles (radixark, 2025) build on top of SGLang, a high-performance open-source inference engine, and focus on delivering a scalable rollout engine to support serverized generation and efficient data collection for large-scale post-training workflows.

However, these systems are primarily optimized for RL throughput rather than production-level speculative decoding, whose goal is end-to-end serving efficiency under strict SLOs. They therefore do not treat the speculator's online learning loop as a first-class closed-loop serving component, nor do they address key deployment challenges such as drafter–verifier mismatch under domain shift/verifier drift and non-disruptive synchronization.

## 3. Unified Training and Inference Framework

We present a unified, production-ready system *Aurora*, that tightly integrates speculative inference and online training into closed loop. Unlike conventional speculative decoding systems that require extensive offline pretraining before deployment, our framework enables **day-0 serving**: a speculator can be deployed *from scratch*, even completely untrained yet rapidly adapted *in situ* on live traffic. This fundamentally changes the speculative decoding from a "train-then-serve" pipeline into a *serve-to-train* flywheel.

### 3.1. System Architecture

As illustrated in Figure 1, our framework consists of two primary decoupled components: an **Inference Server** and a **Training Server**. The Inference Server is responsible for handling user requests. It runs an SGLang (Zheng et al., 2024) speculative decoding engine that uses a target model and a draft model. For each request, the draft model proposes a sequence of tokens, which are then verified in parallel by the target model. The results of both accepted and rejected tokens are streamed to a distributed data buffer. To support EAGLE (Li et al., 2024a;b; 2025) style training, hidden states are also sent to the data buffer.

The Training Server runs asynchronously. It fetches batches of training data from the data buffer and performs gradient updates on a copy of the draft model. Once a new, improved speculator is ready, its weights are asynchronously pushed back to the Inference Server. This update is a *hot-swap*, meaning the Inference Server can begin using the new draft model without downtime or service interruption.

### 3.2. System Implementation

*Aurora* is designed for cost-effective deployment in production environments. We implement batched RPC transfers (using "torch.distributed.rpc" with the TensorPipe backend)

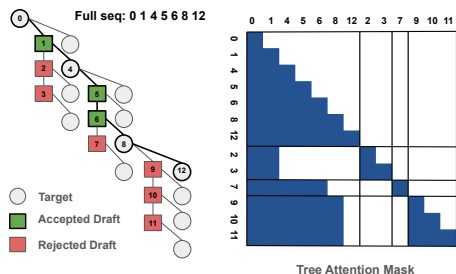

*Figure 2.* Illustration of the Tree Attention mechanism. It enables efficient batched computation over the entire speculative tree, including both accepted (green) and rejected (red) tokens.

with expandable CUDA memory segments to prevent fragmentation and ensure efficient GPU-to-GPU communication. The training loop maintains a thread-safe cache of transmitted data, accumulating samples until reaching the micro-batch size before executing backward passes. Optional checkpointing enables the hot-swapping of updated draft models back to the inference server without downtime, creating a closed-loop adaptation system that responds to distribution shift in real-time production workloads.

**Efficient Tree Attention.** We customize the training mask as the draft model can make mistakes at each step, and training on every verification step individually would be prohibitively expensive. To efficiently process the complex branching structure of the speculative decoding results, we employ a specialized *Tree Attention* mechanism, illustrated in Figure 2. By constructing a custom attention mask that respects the causal structure of the speculative tree, we can process all accepted and rejected branches in a single batched forward and backward pass. This makes it computationally feasible to learn from the entire speculative tree.

This decoupled and asynchronous architecture is the key to our system, *Aurora*. It allows training to run continuously on dedicated resources, leveraging large batch sizes and efficient data processing while inference remains responsive and optimized for low-latency serving. However, making this design work is far from straightforward. The core challenge is orchestrating a reliable closed loop: we need an effective speculator update scheme from on-policy serving traces. Once a new speculator is produced, we then need a careful synchronization policy to roll it into production without perturbing latency or throughput, *i.e.*, choosing an appropriate sync frequency and minimizing sync latency so updates are timely but non-disruptive. Finally, the system must overcome domain shift quickly: as the request mix evolves, the speculator can become stale within hours. To overcome these challenges, we re-formulate the online speculative training as Asynchronous RL problem.

## 3.3. Online Speculator Training as Asynchronous RL

We can view online speculative decoding as an asynchronous reinforcement learning (RL) system to expose the *learning signals* and *systems constraints* that appear when training is embedded directly into live serving. In this formulation, the draft model is the **policy** $\pi$ and the target model plus verifier implement the **environment**. Each speculative step forms a short episode: the policy proposes a tree of candidate continuations, the verifier accepts a prefix, and the outcome provides structured feedback. Accepted tokens correspond to positive reward, while rejected tokens provide zero/negative reward. Maximizing expected return is therefore equivalent to maximizing **acceptance length**, which directly determines decoding speedup.

We realize this view via an asynchronous actor–learner design in AReaL (Fu et al., 2025). As shown in Figure 1, SGLang replicas serve as actors that continuously generate experience from live requests. A distributed buffer aggregates both accepted and rejected branches. A separate multi-GPU learner updates the draft model asynchronously and periodically pushes refreshed weights back to serving replicas. Importantly, we use a lazy synchronization policy: we accept limited staleness to avoid interrupting the inference critical path.

**Learning from Acceptance and Rejection.** Verification yields richer supervision than acceptance-only imitation. We train the draft model with two complementary signals:
*Acceptance loss.* (*imitation*): cross-entropy on accepted tokens, encouraging the draft to reproduce verifier-approved continuations.
*Rejection loss.* (*counterfactual feedback*): rejected branches specify what the policy *should not* propose. With *Discard Sampling*, we apply a KL-based objective (weighted by $\lambda_{\text{discard}}$) that pushes probability mass away from incorrect predictions. Jointly optimizing these objectives produces a denser training signal: acceptance teaches the draft what succeeds under the current traffic, while rejection provides immediate negative feedback on mismatched proposals that would otherwise be discarded. The total loss is a weighted combination of two KL-divergence terms:

$$\mathcal{L} = \mathbb{E}_{x \sim p_{\text{accept}}}[\text{KL}(p_{\text{target}} \| p_{\text{draft}})] \\ + \lambda_{\text{discard}} \, \mathbb{E}_{x \sim p_{\text{discard}}}[\text{KL}(p_{\text{target}} \| p_{\text{draft}})] \quad (3)$$

Here, the first term trains the draft model to mimic the target model on accepted sequences. The second term, our novel *Discard Sampling*, trains the model on rejected sequences, explicitly teaching it to correct its mistakes. We apply top-$k$ filtering to the discarded tokens to focus training on high-probability disagreements, thereby reducing gradient noise.

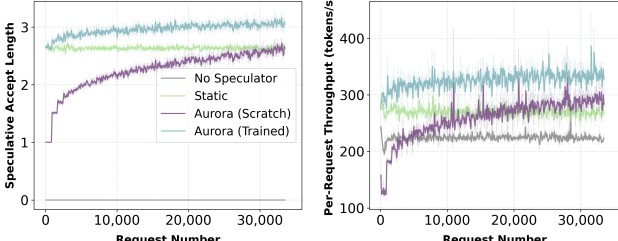

*Figure 3.* **Mixed streams.** Day-0 adaptation of an untrained speculator. (a) Acceptance length starts at one and rapidly increases, converging with the pretrained baseline. (b) Per-request throughput, defined as $(T_{\text{input}} + T_{\text{output}})/t_{\text{request}}$, where $T_{\text{input}}$ and $T_{\text{output}}$ are the input and output token counts and $t_{\text{request}}$ is the end-to-end latency, initially suffers but recovers as the speculator adapts, demonstrating the effectiveness of the serve-to-train flywheel.

# 4. New Feature Study: Day Zero Support Speculator

## 4.1. Online Traffic Simulation

We treat a fixed corpus of prompts as a *stream of inference requests*, rather than a supervised training dataset. Requests are consumed sequentially to simulate live serving traffic, and no ground-truth assistant responses are used.

The stream consists of 40k prompts spanning five domains (Table 2), including mathematical reasoning(Cobbe et al., 2021), text-to-SQL (Yu et al., 2018), code generation (Husain et al., 2019), finance (gbharti, 2026), and general conversational instructions (alespalla, 2026). This composition reflects realistic deployment scenarios where serving traffic exhibits heterogeneous and shifting task distributions.

We evaluate two traffic patterns: (i) **ordered streams**, where requests are grouped by domain to induce abrupt distribution shift, and (ii) **mixed streams**, where prompts are randomly shuffled to approximate stationary traffic. These settings stress different aspects of asynchronous policy updates, allowing us to study the tradeoff between update staleness and serving stability. Each request is processed using EAGLE-3 (Li et al., 2025) speculative decoding, producing both accepted prefixes and rejected branches. These results serve as the sole learning signal for online updates.

**Model and Configuration.** We employ EAGLE-3 (Li et al., 2024b) as our speculative decoding framework. Our target model is Qwen3-8B (Yang et al., 2025), which uses with 4x smaller lm heads (32k vocab size) [2] and is trained on more than 200k dataset. For the day-0 experiment, the draft model is initialized with random weights.

**Methods.** We compare four configurations: (i) an inference system without any speculation, (ii) an inference system

---

[2] https://huggingface.co/Tengyunw/qwen3_8b_eagle3

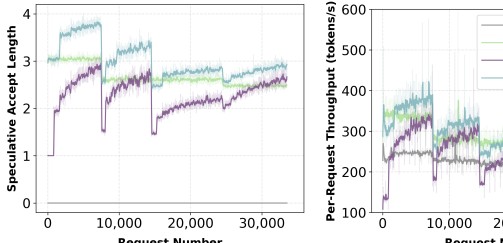

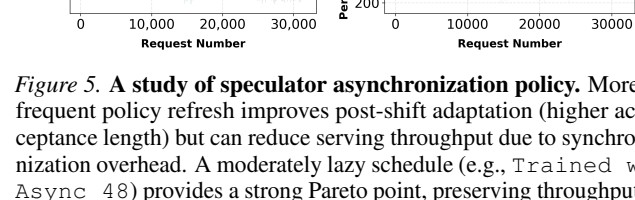

*Figure 4.* **Ordered streams.** Day-0 adaptation of an untrained speculator. (a) Acceptance length starts at one and rapidly increases, converging and sometimes even surpassing the pretrained baseline. (b) Throughput (see definition in Appendix B.1) initially suffers but recovers as the speculator adapts, demonstrating the effectiveness of the serve-to-train flywheel.

*Figure 5.* **A study of speculator asynchronization policy.** More frequent policy refresh improves post-shift adaptation (higher acceptance length) but can reduce serving throughput due to synchronization overhead. A moderately lazy schedule (e.g., `Trained w Async 48`) provides a strong Pareto point, preserving throughput while retaining most of the adaptation benefit.

with a static, pretrained speculator, (iii) *Aurora* (scratch): our unified framework starting from a randomly initialized speculator, and (iv) *Aurora* (trained): our unified framework initialized from the static speculator.

**Results and Implications.** Figure 3 and 4 demonstrate the remarkable effectiveness of our serve-to-train flywheel under two realistic traffic patterns. In the **mixed traffic scenario**, where requests are randomly shuffled to approximate stationary distribution, the system achieves even stronger performance: the acceptance length reaches 3.08 (surpassing both the static baseline at 2.63 and the pretrained-then-finetuned baseline at 2.99), with throughput stabilizing at 302.3 tokens/s. In the **domain shift scenario**, where requests are grouped by domain to induce abrupt distribution changes, the untrained speculator starts at zero acceptance length and converges to 2.46 within approximately 10,000 requests, nearly matching the pretrained baseline's 2.57. Throughput stabilizes at 295.6 tokens/s, competitive with the static speculator's 288.8 tokens/s. Critically, this adaptation happens *during* live serving in both scenarios. The mixed traffic results are particularly striking, showing that online training from scratch can *exceed* the performance of a carefully pretrained speculator. This fundamentally challenges the conventional wisdom that speculative decoding requires extensive offline pretraining. Our unified framework proves that a completely untrained speculator can be deployed on day zero and become production-ready through online adaptation alone, eliminating months of offline training cycles and enabling immediate deployment in novel domains where pretraining data may not exist.

### 4.2. A Trade-off Study of Synchronization Policy

Although the architecture resembles asynchronous RL, the dominant system tradeoff is different: **adaptation speed** versus **serving stability**. In classic async RL, frequent synchronization is beneficial because it reduces policy staleness and improves sample efficiency. In online serving, however,

synchronization has first-order cost: it can disrupt the inference path, invalidate caches, and introduce latency jitter. The key question becomes: *how often can we refresh the draft policy before synchronization overhead negates the serving gains?*

To quantify this, we sweep the policy update interval from aggressive (every 48 requests) to lazy (every 1600 requests), under an identical request stream with an abrupt domain shift. Figure 5 shows a clear tradeoff. Aggressive updates recover acceptance length faster, but suffer the lowest throughput due to frequent weight synchronization. In contrast, a moderately lazy schedule (every 80 requests) achieves nearly the same acceptance recovery while delivering the best overall throughput, even surpassing the baseline.

## 5. Speculative Algorithm Exploration

This section studies which *training objectives* and *online update strategies* matter most for speculative decoding under realistic serving dynamics. Across settings, we find a consistent pattern: once the speculator is trained *on-policy* and kept aligned with the current traffic, simple token-level objectives capture most of the attainable gains, while more complex supervision (e.g., tree-structured losses) provides diminishing returns under domain shift.

### 5.1. Baselines and Variants

We compare a static deployed speculator against a spectrum of online adaptation methods (all within the same training–serving loop): (1) *Frozen Draft (Static Baseline).* A pretrained speculator is deployed for speculative decoding with *no* online updates, representing the standard non-adaptive setting. (2) *Aurora (FKL).* Online fine-tuning using only *accepted* tokens from verification with a forward KL objective. (3) *Aurora (RKL).* Online fine-tuning using only *accepted* tokens from verification with a reverse KL objective. (4) *Aurora (RKL + NTP).* Method (2) augmented with an auxiliary

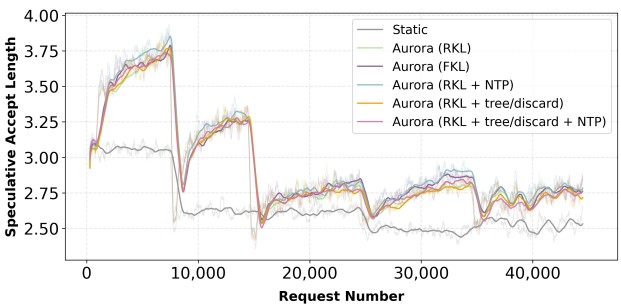

*Figure 6.* Moving-average speculative decoding acceptance length over inference requests for Qwen-8B-Instruct. Online fine-tuning substantially improves accept length over the frozen baseline. Training with different strategies yields only marginal differences under domain shift.

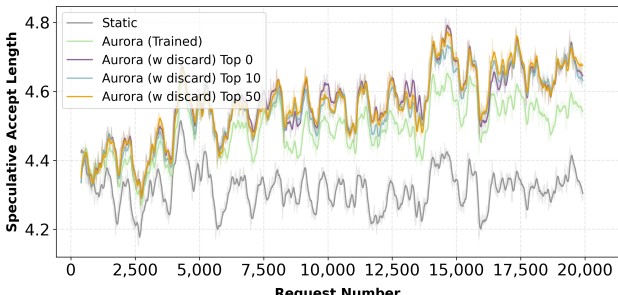

*Figure 7.* Moving-average speculative decoding accept length over inference requests for Llama-3.1-8B-Instruct (Dubey et al., 2024) for coding with lookahead = 10. Training on discarded tokens provides additional gains on the Llama model, but different discard strategies (top-$k$ = 0, 10, 50) yield only marginal differences, indicating that the top-$k$ strategy saves memory while preserving performance.

next-token prediction loss on accepted tokens to strengthen the training signal. (5) *Aurora (w discard).* Extends training to include tokens from *rejected* speculative branches, allowing the draft to learn directly from its mistakes.

We evaluate these methods across multiple models and dataset configurations using two metrics: *Speculative Acceptance Length* (the average number of draft tokens accepted per verification step) and *Per-Request Throughput* (end-to-end throughput/latency improvement).

**Online finetuning provides substantial and stable improvements.** Across all settings, online finetuning consistently achieves higher throughput than the Static Baseline, with the gain increasing as more tokens are used for training. In contrast, incorporating NTP and discarded tokens yields only marginal additional improvements.

**Marginal benefits of discard tokens under domain shift.** Across all settings, the dominant gains come from *closing the loop*: on-policy online updates substantially improve acceptance length and stabilize throughput after traffic shifts. In contrast, adding discarded tokens provides only incremental improvements and often does not separate cleanly from the simplest online objectives.
Figure 6 illustrates this behavior: the acceptance length exhibits clear regime changes consistent with the distribution shift; the frozen baseline degrades and remains lowest, whereas all online variants follow a higher plateau and recover after each shift. Within the online cluster, tree/discard variants largely overlap with RKL(+NTP), indicating limited headroom beyond token-level adaptation in this online setting.

**When does discarded tokens help?** In previous experiments with lookahead = 5, the pretrained speculator already achieved a relatively high acceptance length (e.g., ~3.6), leaving limited headroom for more expressive supervision

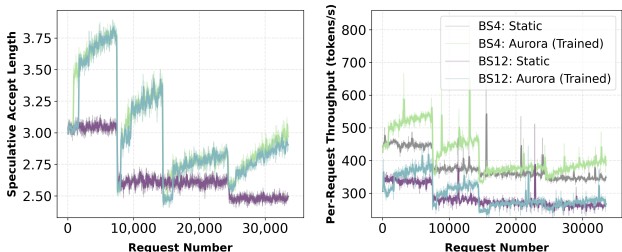

*Figure 8.* Aurora increases speculative accept length and boosts throughput, with larger speedups at smaller batch sizes. Here, BS4 and BS12 denote 4 and 12 concurrent requests per batch, respectively.

to matter. This motivates a simple hypothesis: *discarded tokens become more useful when the serving setting demands longer accepted prefixes*. To test this, we increase the lookahead to 10. As shown in Figure 7, under this more aggressive lookahead, learning from discarded tokens provides a noticeably stronger training signal and yields greater gains over accept-only fine-tuning.

**How does *Aurora* perform with different batch size?** Reducing the batch size from 12 to 4 (Figure 8), *Aurora* sustains a higher acceptance length than the static drafter across traffic phases (with drops at shifts followed by re-adaptation), while the baseline stays flat. The accepted prefix converts into higher tokens/s, but the relative uplift is larger at batch size 4: at small batches, baseline decoding is dominated by per-step overhead and poor hardware utilization, so longer accepted prefixes yield a large proportional win; at larger batches the target model is already well amortized and speculative overhead (drafting, verification, synchronization) becomes a larger fraction of the pipeline, shrinking the net speedup even as acceptance improves.

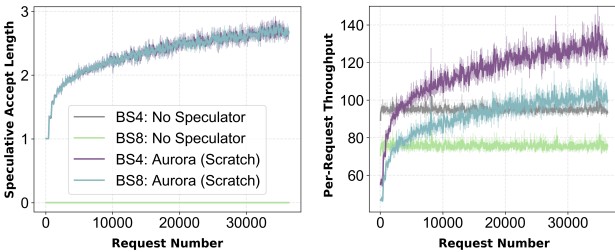

*Figure 9.* **MiniMax M2.1.** Left: accepted draft length over time. Right: per-request throughput over time. *Aurora* (Scratch) increases acceptance length to 2.8 and translates it into $1.45\times$ throughput (BS4) gains over the no-speculation baseline.

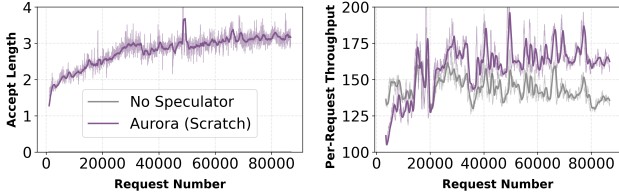

*Figure 10.* **Qwen3-Coder-Next.** Left: accepted draft length over time. Right: per-request throughput over time. We discard the first 1,000 warm-up steps, since hybrid deployments exhibit transient throughput instability during initialization. Despite this variability, *Aurora* (SCRATCH) raises the mean accepted draft length to 3 and delivers a $1.21\times$ throughput improvement over the no-speculation baseline (averaged over the final 10k steps). All results use a batch size of 8.

## 6. Scalability on Frontier Open-sourced Models

We evaluate scalability on two recent open-source frontier models, showing that *Aurora* keeps delivering stable gains as model size, architectural complexity, and context length grow. **MiniMax M2.1** (Chen et al., 2025; MiniMaxAI, 2025) (released 2025/12/23) is a 229B-parameter decoder-only Transformer-MoE with 62 layers and a 256-expert router (top-8 per token) and a 196K native context window. **Qwen3-Coder-Next** (Qwen Team, 2026) (released 2026/02/02) is an MoE with 80B total / 3B active parameters, 48 layers, and a 262K context length; its hybrid backbone interleaves Gated DeltaNet (Yang et al., 2024) and Gated Attention blocks paired with MoE layers (512 experts, top-10), and speculative decoding for such hybrid models was first proposed in (Wang et al., 2024).

We serve MiniMax-M2.1 on 4×H200 GPUs with tensor parallelism (TP)=4 in `FP8`, allocating one extra GPU for online speculator tuning. *Aurora* (SCRATCH) raises the mean accepted draft length to 2.8 and converts it into about $1.45\times$ throughput (Figure 9), remaining effective even under large-expert routing. Qwen3-Coder-Next is served on 4×H200 GPUs with TP=4 and expert parallelism (EP)=4 in

*Table 1.* **End-to-end throughput under varying batch sizes.** TPS = tokens-per-second (see definition in Appendix B.1). MiniMax M2.1 (FP8) uses lookahead 4; Qwen3-Coder-Next (FP8) uses lookahead 5. Full results in Tables 4,5.

| BS | Config | Mean | P50 | P05 | P95 | Speedup | Acc Len |
|---|---|---|---|---|---|---|---|
| *MiniMax M2.1 (FP8)* | | | | | | | |
| 1 | w/o spec | 134.9 | 136.4 | 130.6 | 136.9 | – | – |
| | w/ spec | 211.8 | 210.6 | 163.1 | 270.3 | **1.57×** | 2.62 |
| 8 | w/o spec | 79.0 | 78.7 | 73.7 | 85.1 | – | – |
| | w/ spec | 107.1 | 104.5 | 79.9 | 137.1 | **1.36×** | 2.62 |
| 16 | w/o spec | 64.5 | 63.7 | 58.9 | 72.3 | – | – |
| | w/ spec | 83.1 | 82.9 | 60.9 | 112.0 | **1.29×** | 2.62 |
| 32 | w/o spec | 53.5 | 52.9 | 47.1 | 67.1 | – | – |
| | w/ spec | 67.1 | 64.7 | 44.0 | 100.5 | **1.25×** | 2.62 |
| *Qwen3-Coder-Next (FP8)* | | | | | | | |
| 1 | w/o spec | 176.4 | 178.0 | 172.3 | 178.4 | – | – |
| | w/ spec | 265.7 | 264.8 | 208.7 | 320.5 | **1.51×** | 3.06 |
| 8 | w/o spec | 119.8 | 121.5 | 104.8 | 134.6 | – | – |
| | w/ spec | 146.3 | 143.5 | 109.6 | 189.5 | **1.23×** | 3.07 |
| 16 | w/o spec | 99.6 | 102.1 | 74.5 | 119.2 | – | – |
| | w/ spec | 107.6 | 103.7 | 75.7 | 156.6 | **1.09×** | 3.06 |

`FP8` (one extra tuning GPU); *Aurora* consistently increases acceptance beyond 3 and yields $1.23\times$ throughput on this hybrid TP/EP deployment (Figure 10).

**Performance with different batch size.** Using our final *Aurora* checkpoints on held-out data (Table 1), *Aurora* improves throughput across batch sizes, with the largest gains at small-to-moderate batches: from $1.57\times$ down to $1.25\times$ for MiniMax M2.1, and up to $1.51\times$ (batch size 1) for Qwen3-Coder-Next. Returns diminish as batch size grows; at batch size 32 verification overhead dominates and speculative decoding can be slightly slower than the baseline. Future work includes faster speculative decoding for hybrid models, potentially via multistep kernels (Wang et al., 2024).

## 7. Conclusion

We presented *Aurora*, a unified training-serving system that recasts speculative decoding as a joint learning-and-serving problem. By connecting an SGLang inference server to an asynchronous training server via GPU-aware RPC, *Aurora* continuously adapts the draft model on-policy under live traffic, closing the training-serving mismatch of conventional two-stage pipelines. Across models, simple online fine-tuning captures most attainable gains, lazy synchronization best balances adaptation with stability, and day-0 deployment from scratch is practical: an untrained speculator becomes competitive within thousands of requests, removing the offline pretraining bottleneck for onboarding new models.

## Impact Statement

This paper presents *Aurora*, a unified training–serving system that continuously adapts a speculative-decoding draft model from live inference traffic. The primary impact is improved efficiency of large language model (LLM) serving: by raising token acceptance under shifting workloads, *Aurora* reduces the number of expensive target-model decode steps, which can lower the latency, energy consumption, and monetary cost of deploying LLMs. Because the speculator is verified against the unchanged target model, our approach preserves the target model's output distribution and therefore does not alter the quality or safety properties of the served model.

The main consideration specific to our setting is that *Aurora* trains on live serving traffic. Deployments must therefore ensure that any user data used to update the speculator is handled in accordance with applicable privacy policies and data-retention agreements; the bounded on-policy buffer we use stores only the signals needed for training and can be configured to respect such constraints. More broadly, cheaper inference may increase aggregate LLM usage, which carries the standard dual-use and environmental considerations associated with the wider deployment of generative models. We do not foresee additional societal consequences that are unique to this work beyond those already associated with accelerating LLM inference.

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

# A. Technical Details for *Aurora* System

A key technical contribution of our work is the design of an asynchronous, decoupled training architecture that enables inference-time training without disrupting the production inference pipeline. We propose a GPU-aware Remote Procedure Call (RPC) based system that bridges the inference server and training server, enabling efficient GPU-to-GPU data transfer in the background across nodes while maintaining low inference latency.

Our architecture consists of two independent components: (1) a **production inference server** that serves real user requests with speculative decoding, and (2) a **passive training server** that continuously updates the draft model based on incoming inference data. Unlike prior approaches that either require offline training or duplicate model loading, our system achieves online training with minimal memory overhead through a carefully designed RPC communication channel.

## A.1. Zero-Copy Target Model Design

A critical insight of our approach is that during inference-time training, we can **eliminate the need to load a duplicate target model** on the training server. Traditional speculative decoding training requires computing target model hidden states and logits, which necessitates loading the full target model (e.g., 8B parameters $\approx$ 16GB in FP16). Our RPC-based design transmits pre-computed target information directly from the inference server:

$$\mathcal{D}_{\text{RPC}} = \{(h_t^{(l)}, \ell_t, x_{\text{in}}, y_{\text{out}}, \mathcal{R})\} \tag{4}$$

where $h_t^{(l)} \in \mathbb{R}^{T \times 3d}$ represents concatenated hidden states from three strategically selected target model layers (early, middle, late), $\ell_t \in \mathbb{R}^{T \times V}$ denotes the output logits, $x_{\text{in}}$ and $y_{\text{out}}$ are input and output token sequences, and $\mathcal{R}$ encodes rejection trajectory information. This design achieves:

- **Training memory efficiency**: Training nodes only load the draft model ($\sim$1B), not the target model (8B to 70B)

- **Hot swap, minimal Inference Disruption**: Training runs on separate hardware, resulting in minimal serving overhead

- **Horizontal Scaling**: Multiple inference servers can feed a single training server

## A.2. Inference-Side Memory Overhead

Although our design eliminates the target model from the training server, it introduces a modest memory overhead on the inference server to buffer auxiliary data before RPC transmission. For each request with sequence length $T = T_{\text{prompt}} + T_{\text{output}}$, the server must temporarily store: (1) auxiliary hidden states from target layers, consuming $T \times 3d \times 2$ bytes in BF16 models, and (2) target logits for output tokens, consuming $T_{\text{output}} \times V \times 2$ bytes. For a batch of $B$ requests, the total auxiliary memory is:

$$M_{\text{aux}} = B \times \left( T \times 3d \times 2 + T_{\text{output}} \times V \times 2 \right) \tag{5}$$

To mitigate the logit bottleneck, we exploit the fact that most draft models have significantly smaller vocabularies than their target counterparts (e.g., 32K vs 128K for Llama3 models and 32K vs 152K for Qwen3 models), so we only need to transmit logits over the draft vocabulary—a $4\times$ to $5\times$ reduction. We can further apply top-$K$ logits filtering (e.g., $K=1024$) to retain only the most informative target logits, reducing per-token logit storage from 256 KB to 2 KB (a $128\times$ reduction), making the hidden states the dominant cost.

## A.3. Multi-Server Aggregation

Multiple inference servers can simultaneously send data to a single training server. Each server is assigned a unique RPC rank, and the training server aggregates data through thread-safe LRU caching:

$$\text{RPC World} = \{\underbrace{S_1, S_2, \ldots, S_N}_{\text{Inference Servers}}, \underbrace{T}_{\text{Training Server}}\} \tag{6}$$

For heterogeneous cluster configurations, explicit GPU device mapping enables flexible deployment where inference servers use GPUs 0-3 for tensor parallelism while the training server uses GPUs on a completely separate node.

## A.4. Compatibility with Disaggregated Serving

Our decoupled architecture naturally extends disaggregated serving frameworks such as Mooncake (Qin et al., 2024) and DistServe (Zhong et al., 2024), which already separate prefill and decoding onto distinct node pools with cross-node GPU transfer infrastructure. Our training server acts as a third disaggregated role, receiving hidden states and logits over the same communication fabric (e.g., RDMA) without requiring a separate data path. This makes our inference-time training pipeline deployable atop any serving backend—monolithic or disaggregated—without changes to the training logic.

## A.5. The Generalization of *Aurora* System

We note that *Aurora* can be used not only for online serving, but also for traditional speculator training as well by simply simulating the online processing pipeline using a pre-existing corpus.

For speculator training, users typically encounter two cases: 1) When a generated corpus from the target model is unavailable, we can train the speculator by collecting online data using pre-collected prompts. 2) When a target model generated corpus is available, we can simply run the inference component in a prefill-only mode to efficiently produce the activations required for speculator training.

Even when viewed as a traditional speculator training framework, *Aurora* still has advantages: training and inference can run in parallel, making it more efficient than previous systems, and users can immediately measure speedups, avoiding the need to estimate throughput improvements from the acceptance rate or risk misestimating throughput due to training–serving mismatch. Thus, we believe that *Aurora* can be efficiently adapted to different user scenarios.

## A.6. Dataset Description

*Table 2.* Multi-domain dataset composition for inference-time training experiments.

| Domain | Samples |
|---|---|
| Mathematical Reasoning | 7k |
| Text-to-SQL | 7k |
| Code | 10k |
| Finance | 10k |
| Conversational Instructions | 10k |
| **Total** | **44k** |

**Data composition** We curate a mixed-domain prompt stream that spans mathematical reasoning, structured query generation, code-centric tasks, finance-domain instructions, and general conversational prompts. Specifically, Mathematical Reasoning is sampled from GSM8K (Cobbe et al., 2021), which contains multi-step grade-school math word problems. Text-to-SQL is from Spider (Yu et al., 2018), where the model generates executable SQL queries from natural-language questions given a database schema. Code data is sampled from CodeSearchNet (Husain et al., 2019), which covers code and associated natural-language descriptions across multiple programming languages. Finance data is sampled from Finance-Alpaca (gbharti, 2026), consisting of finance-related instruction follow-up prompts with domain-specific terminology and reasoning. The conversational Instructions come from `chatbot_instruction_prompts` (alespalla, 2026), which contains general assistant-style prompts for open-ended dialog.

**Dataset representativeness.** Our dataset covers diverse token-level distributions, which directly determine the draft–target matching patterns and thus the effectiveness of speculative decoding under distribution shift. Specifically, GSM8K features numerical symbols and multi-step reasoning traces with relatively consistent answer structures; Spider requires strongly structured SQL generation with strict syntax constraints and frequent keywords/punctuation; CodeSearchNet contains code-centric outputs with high-frequency programming tokens (e.g., indentation and operators), long-tail identifiers, and strong local repetition; Finance-Alpaca introduces domain-specific terminology (e.g., tickers and macroeconomic concepts) with explanation-style responses; and `chatbot_instruction_prompts` represents open-ended assistant conversations with more diverse and less constrained language. This diversity allows us to reliably induce distribution shifts that lead to a noticeable drop in acceptance rate, and to evaluate whether online speculator training can recover performance over time.

Moreover, the dataset captures two common output regimes in real-world assistant traffic: (i) long-form natural language

responses (Finance and Conversational Instructions), and (ii) format-constrained structured generation (Text-to-SQL and Code). Compared to evaluating solely on standard QA prompts, these regimes better reflect scenarios where speculative decoding benefits from predictable structural fragments (e.g., `SELECT ... FROM ... WHERE ...`) and recurring templates that appear frequently in structured outputs. Overall, our domain composition provides an intuitive and realistic shift setting, e.g., traffic transitioning from reasoning-heavy queries to code/SQL/finance/conversational workloads.

## B. Training Details

### B.1. Infrastructure and Efficiency

To enable online training without degrading inference throughput, we employ a split-GPU architecture with RPC-based data transfer:

**GPU Allocation:** We place the target model on GPUs 0–3 for inference, while training the draft model on a separate GPU to avoid contention between serving and learning. All Qwen-8B and Llama-8B experiments are conducted on H100 GPUs, whereas MiniMax M2.1 and Qwen3-Coder-Next experiments are conducted on H200 GPUs.

**Data Transfer:** Hidden states and verification outcomes are transmitted via RPC from the inference server to the training process, with storage buffers co-located on GPU 1 to avoid redundant memory copies.

**Asynchronous Updates:** Gradient updates proceed asynchronously with respect to inference, with updates triggered every 100 inference requests. This update interval allows the training buffer to accumulate diverse samples before each optimization step, balancing training responsiveness with computational efficiency while ensuring that serving latency remains unaffected by training overhead.

**Throughput Metrics.** Throughout this paper, we report *per-request throughput* measured in tokens per second (tokens/s), defined as the total number of tokens processed for a single request divided by the end-to-end latency:

$$\text{Throughput} = \frac{T_{\text{input}} + T_{\text{output}}}{t_{\text{request}}}, \tag{7}$$

where $T_{\text{input}}$ is the number of input (prompt) tokens, $T_{\text{output}}$ is the number of generated output tokens, and $t_{\text{request}}$ is the wall-clock time from request submission to response completion. This metric captures the effective token processing rate experienced by individual users and directly reflects the latency improvements enabled by speculative decoding. Unlike aggregate system throughput (which measures total tokens across concurrent requests), per-request throughput isolates the speedup benefit for each query, making it the appropriate metric for evaluating user-perceived performance gains under different speculative decoding configurations.

### B.2. Optimization and Hyperparameters

All online training configurations share a common optimization setup designed for stable streaming updates under inference workload constraints:

**Optimizer:** AdamW with weight decay $\beta = 0.0$ and gradient clipping at max norm 0.5.

**Learning Rate:** We use a constant learning rate of $\eta = 1 \times 10^{-5}$ for finetuning and $\eta = 1 \times 10^{-4}$ for training from scratch after a brief linear warmup over 400 steps (0.05% of total training). This constant schedule is critical for test-time training scenarios where the model must maintain plasticity for continuous adaptation without forgetting (Sun et al., 2020).

**Batch Configuration:** Global batch size of 8 for draft model training, with micro-batch sizes of 1 for draft model forward passes and 4 for target model verification. These asymmetric batch sizes balance training stability with inference throughput requirements.

**Context Window:** Maximum sequence length of 2,048 tokens with a test-time training (TTT) length of 5 tokens, defining the local context window for each gradient update.

**Mixed Precision:** BF16 computation with FP32 master weights and gradients cast to FP32 before optimization, following best practices for training stability (Micikevicius et al., 2017).

**Loss Function:** Reverse KL divergence $D_{\text{KL}}(p_{\text{target}}\|p_{\text{draft}})$ encourages the draft model to cover the target model's distribution, minimizing false rejections during speculative verification.

*Table 3.* Shared hyperparameters across all configurations. Constant LR maintains plasticity for online learning.

| Parameter | Value |
|---|---|
| *Training* | |
| Learning Rate | $10^{-5}/10^{-4}$ |
| Warmup Steps | 400 |
| Global Batch Size | 8 |
| Draft/Target Micro-batch | 1 / 4 |
| Optimizer | AdamW |
| Weight Decay | 0.0 |
| Gradient Clipping | 0.5 |
| Precision | BF16 + FP32 |
| *Sequence & Loss* | |
| Max Sequence Length | 2,048 |
| TTT Length | 5 |
| Top-$k$ Filter (discard) | 10 |
| Discard Loss Weights ($\lambda$) | 1.0 |
| Divergence | Reverse KL |
| *Speculative Decoding Inference* | |
| Speculative Steps | 5 |
| Top-$k$ Sampling | 1 |
| Draft Tokens per Step | 6 |

## B.3. Additional Results

*Table 4.* **MimiMax M2.1 (BF16): end-to-end throughput under varying batch size and lookahead.** We report tokens-per-second (TPS, defined in Appendix B.1) and speedup relative to the no-speculation baseline.

| BS | Config | Mean TPS | P50 TPS | P05 TPS | P95 TPS | Count | Speedup | Acc Len |
|---|---|---|---|---|---|---|---|---|
| 1 | w/o spec | 134.9 | 136.4 | 130.6 | 136.9 | 257 | – | – |
| | lookahead 3 | 213.0 | 213.7 | 169.8 | 256.3 | 257 | **1.58**× | 2.42 |
| | lookahead 4 | 211.8 | 210.6 | 163.1 | 270.3 | 257 | **1.57**× | 2.62 |
| 8 | w/o spec | 79.0 | 78.7 | 73.7 | 85.1 | 257 | – | – |
| | lookahead 3 | 106.5 | 105.2 | 84.0 | 134.8 | 257 | **1.35**× | 2.43 |
| | lookahead 4 | 107.1 | 104.5 | 79.9 | 137.1 | 257 | **1.36**× | 2.62 |
| | lookahead 5 | 106.6 | 104.8 | 79.3 | 140.9 | 257 | **1.35**× | 2.70 |
| 16 | w/o spec | 64.5 | 63.7 | 58.9 | 72.3 | 257 | – | – |
| | lookahead 3 | 83.2 | 81.4 | 62.2 | 110.3 | 257 | **1.29**× | 2.43 |
| | lookahead 4 | 83.1 | 82.9 | 60.9 | 112.0 | 257 | **1.29**× | 2.62 |
| | lookahead 5 | 82.6 | 81.0 | 58.1 | 116.1 | 257 | **1.28**× | 2.69 |
| 32 | w/o spec | 53.5 | 52.9 | 47.1 | 67.1 | 257 | – | – |
| | lookahead 3 | 67.1 | 64.9 | 45.2 | 97.8 | 257 | **1.25**× | 2.44 |
| | lookahead 4 | 67.1 | 64.7 | 44.0 | 100.5 | 257 | **1.25**× | 2.62 |
| | lookahead 5 | 67.3 | 64.9 | 45.2 | 99.7 | 257 | **1.26**× | 2.71 |

*Table 5.* **Qwen-Coder-Next: end-to-end throughput under varying batch size and lookahead.** We report tokens-per-second (TPS) statistics and speedup relative to the no-speculation baseline.

| BS | Config | MeanTPS | P50 TPS | P05 TPS | P95 TPS | Speedup (Mean) | Acc Len |
|---|---|---|---|---|---|---|---|
| 1 | w/o spec | 176.4 | 178.0 | 172.3 | 178.4 | – | – |
| | lookahead 3 | 252.1 | 254.8 | 208.8 | 291.6 | 1.43× | 2.67 |
| | lookahead 4 | 263.1 | 264.0 | 211.8 | 312.7 | 1.49× | 2.91 |
| | lookahead 5 | 265.7 | 264.8 | 208.7 | 320.5 | **1.51×** | 3.06 |
| 8 | w/o spec | 119.8 | 121.5 | 104.8 | 134.6 | – | – |
| | lookahead 3 | 141.0 | 138.9 | 110.4 | 178.5 | 1.18× | 2.67 |
| | lookahead 4 | 142.5 | 141.2 | 110.3 | 181.6 | 1.19× | 2.91 |
| | lookahead 5 | 146.3 | 143.5 | 109.6 | 189.5 | **1.23×** | 3.07 |
| 16 | w/o spec | 99.6 | 102.1 | 74.5 | 119.2 | – | – |
| | lookahead 3 | 104.0 | 100.5 | 75.6 | 151.9 | 1.04× | 2.67 |
| | lookahead 4 | 105.6 | 101.1 | 77.5 | 149.7 | 1.06× | 2.92 |
| | lookahead 5 | 107.6 | 103.7 | 75.7 | 156.6 | **1.09×** | 3.06 |
| 32 | w/o spec | 85.0 | 88.7 | 54.5 | 104.5 | – | – |
| | lookahead 3 | 78.9 | 72.8 | 53.0 | 122.3 | 0.93× | 2.68 |
| | lookahead 4 | 79.5 | 73.7 | 52.9 | 124.7 | 0.94× | 2.91 |
| | lookahead 5 | 80.3 | 72.6 | 52.8 | 130.7 | 0.94× | 3.06 |

