# OpenReview forum: "When RL Meets Adaptive Speculative Training:  A Unified Training-Serving System"
_ICML.cc/2026/Conference — ICML 2026 regular_

### Official Review · Reviewer_9hFM · 2026-02-21

**Soundness:** 3
**Presentation:** 3
**Significance:** 2
**Originality:** 3
**Overall Recommendation:** 4
**Confidence:** 3

**Summary:**

This paper introduces Aurora, a system designed to update the drafter model for speculative decoding concurrently with LLM serving. Traditional speculative decoding typically employs a static drafter model, which often leads to a decline in the acceptance rate. Aurora leverages reinforcement learning to optimize the drafter model using real-time inference data, ensuring that the drafter maintains a consistently high acceptance rate.
Experimental results demonstrate that Aurora significantly enhances the effectiveness of speculative decoding, gaining substantial speedups in end-to-end serving.

**Compliance With Llm Reviewing Policy:**

Affirmed.

**Final Justification:**

I have read the authors' rebuttal, which fully addressed my main concerns. The newly provided latency measurements confirm that Aurora's runtime overhead introduces negligible P99 tail latency increase compared to the baseline. The missing baselines are also reasonably explained by the gap between simulation-level implementations and production inference engines.

I maintain my score of 4 (Weak Accept). The paper presents an original and well-motivated approach to a real limitation of speculative decoding. The RL-based online training framework is technically sound, and the empirical study on factors influencing acceptance rate is thorough. My remaining reservation is on significance: the practical impact depends on how frequently real-world serving traffic exhibits the kind of distribution shift that necessitates continuous drafter adaptation, and the paper could benefit from further analysis on this point.

**Key Questions For Authors:**

1. If the RL environment for training the drafter model is consistent with the deployment environment, and the drafter is updated timely (which is much less costly than training the base model), the overhead seems manageable. Why is it necessary to perform real-time training during online deployment? Furthermore, does training on offline datasets lead to a significant drop in accuracy or acceptance rate?
2. The authors introduce several existing works related to online speculative decoding in the Background section. Why were these not included as baselines in the experimental evaluation?
3. There is a lack of experimental analysis regarding the impact of background training tasks on inference performance. Does the concurrent training task introduce significant latency jitter or tail latency (P99) for the inference requests?

**Limitations:**

yes

**Strengths And Weaknesses:**

Strengths
1. The real-time updates to the drafter model effectively boost the acceptance rate, which directly translates into improved system throughput.
2. The paper provides a solid and rigorous empirical study on the various factors that influence the acceptance rate in speculative decoding.

Weaknesses
1. The computational overhead of training a drafter model is generally marginal compared to the resources required for training the target (base) model.
2. In Aurora, training tasks are co-located on the same GPUs used for inference to maximize resource utilization. However, under high-load scenarios, the competition for compute resources and memory bandwidth between training and inference may negatively impact inference latency.

---

> ### Author Rebuttal · Authors · 2026-03-31
>
> We thank the reviewer for the thoughtful feedback and respond below.
>
> ### W1: Training Overhead and GPU Co-location
>
> We clarify an important misunderstanding: training and inference run on separate, non-overlapping GPUs. As described in Appendix A (Section A.1): "Training nodes only load the draft model (\~1B), not the target model (8B to 70B)." In our configurations, the target model runs inference on dedicated GPUs (e.g., GPUs 0–3 with TP=4), while the draft model trains on a separate GPU (e.g., GPU 4\) or on a completely separate node. There is no resource contention between training and inference.
>
> This is further demonstrated by our multi-node examples, where the inference server and training server run on physically separate machines connected via RPC.
>
> ### W2: Why Real-Time Training During Online Deployment?
>
> Two key reasons:
>
> 1. **Distribution shift adaptation.** Offline training on a fixed dataset cannot adapt to changing traffic patterns. Figures 3–4 demonstrate this clearly: the static (offline-trained) speculator freezes at suboptimal acceptance lengths under domain shift, while Aurora continuously adapts. In the mixed traffic scenario (Figure 3), Aurora's online-trained speculator even *surpasses* the pretrained-then-finetuned baseline (3.08 vs. 2.99 acceptance length).
>
> 2. **Day-0 serving.** Online training enables deploying a speculator from random initialization on day one — no offline pretraining needed. Figures 3–4 show that an untrained speculator rapidly reaches competitive acceptance rates within thousands of requests. This eliminates the weeks-long offline training cycles typically required before speculative decoding can be deployed on a new model.
>
> ### W3: Does Offline Training Lead to Acceptance Drop?
>
> Yes. Figures 3 and 4 demonstrate this directly. The Static baseline is an offline-trained speculator using the standard SpecForge pipeline. Under domain shift (Figure 4, ordered streams), its acceptance length stagnates while Aurora continuously improves. Under mixed traffic (Figure 3), Aurora surpasses even the pretrained baseline, confirming that on-policy online training produces a better-adapted speculator than offline training on a fixed corpus.
>
> ### W4: Online Speculative Decoding Baselines
>
> The existing [online speculative decoding](https://github.com/LiuXiaoxuanPKU/OSD/issues/10) focuses on simulation-level implementations rather than integration with real inference engines such as SGLang or vLLM. As a result, its performance is significantly lower than that of production-grade inference engines like SGLang.
>
> ### W5: Does the concurrent training task introduce significant latency jitter or tail latency.
>
> We report the overhead of hidden-state/logit transformation and weight synchronization in our measurements. In this experiment, the speculator weights are kept unchanged (Aurora speculator frozen), so the measured overhead reflects only the runtime cost of transformation and synchronization, without any effect from weight updates.
>
> | Generation Length | System | Throughput (tok/s) | Latency P50 (ms) | Latency P99 (ms) | TPOT P50 (ms) | TPOT P99 (ms) |
> |:--|:--|--:|--:|--:|--:|--:|
> | 1024 | Baseline (SGLang + EAGLE-3) | 359.4 | 2840.4 | 3256.1 | 2.78 | 3.18 |
> | 1024 | Aurora (speculator frozen) | 354.7 | 2876.2 | 3303.0 | 2.82 | 3.23 |
> | 1024 | Aurora (speculator trained) | 499.8 | 2048.2 | 2409.0 | 2.01 | 2.35 |
> | 2048 | Baseline (SGLang + EAGLE-3) | 370.4 | 4942.6 | 6256.5 | 2.70 | 3.06 |
> | 2048 | Aurora (speculator frozen) | 360.6 | 4985.2 | 6399.6 | 2.78 | 3.15 |
> | 2048 | Aurora (speculator trained) | 504.1 | 3662.0 |4719.3 | 1.97 | 2.30|
>
> As shown in the table, Aurora’s P99 tail latency with the speculator frozen remains similar to that of standard inference, suggesting that it does not introduce significant additional overhead.

---

> > ### Author Rebuttal · Reviewer_9hFM · 2026-04-01
> >
> > Thank you for the rebuttal. My concerns have been addressed. I will maintain my positive score.

---

### Official Review · Reviewer_Xu55 · 2026-03-06

**Soundness:** 2
**Presentation:** 2
**Significance:** 3
**Originality:** 3
**Overall Recommendation:** 4
**Confidence:** 4

**Summary:**

This paper presents Aurora, a unified training-serving framework for speculative decoding that updates the speculator online using feedback from live serving.  Aurora closes the loop between inference and training by collecting accepted and rejected speculative branches during serving and using them to continuously improve the draft model. The paper introduces training mechanisms for learning from both accepted and discarded branches, together with a Tree Attention scheme for efficient full-tree supervision. Experiments show that Aurora supports day-0 deployment from scratch, adapts effectively under mixed and shifting traffic, and improves speculative decoding performance through continuous online adaptation.

**Compliance With Llm Reviewing Policy:**

Affirmed.

**Final Justification:**

During my initial review, I was intrigued by the research question addressed in this paper. The authors propose a simple yet effective approach that combines speculative encoding with online training, demonstrating its effectiveness in day0 deployment and adapting to traffic shifts. My primary concern, however, was the paper’s theoretical contribution, which seemed to rely more on engineering intuition than rigorous mathematical reasoning. Another concern was the scalability of the approach. The authors have substantially addressed my concerns in their rebuttal, so I am maintaining a weak accept.

**Key Questions For Authors:**

Please refer to Weaknesses 1-4.

**Limitations:**

While the paper discusses practical tradeoffs such as synchronization overhead and adaptation stability, it provides only limited discussion of the generalization scope of the proposed framework. In particular, although Aurora is claimed to be largely model-agnostic and mechanism-agnostic, the empirical support for this claim remains limited. A more complete limitations discussion should acknowledge that the current results are demonstrated under a relatively narrow range of model, serving, and traffic configurations.

**Strengths And Weaknesses:**

**Strengths**
1. The paper clearly identifies an important and practical problem in speculative decoding: the training-serving mismatch caused by static speculators under changing live traffic, and suggests an important paradigm shift: speculative decoding could be treated as a continuously adapting system, not just a static acceleration module.

2. The proposed Aurora presents a fairly complete framework, including the unified online training-serving loop, asynchronous hot-swapping, learning from accepted and discarded branches, and Tree Attention for full-tree supervision. The design is simple and well integrated.

3. The experiments show that Aurora can mitigate mismatch effectively and support day-0 serving from scratch, which is a compelling practical result.

**Weaknesses**
1. Theoretical contributions are limited. The paper demonstrates that online adaptation works, but provides limited analytical understanding of why mismatch arises, or how to choose update frequency in a principled way (how to model the adaptation–stability tradeoff).

2. Tree Attention is sensible, but it likely introduces nontrivial memory and backward-pass overhead. The paper should quantify its standalone cost in terms of training time, memory usage, and scaling with tree size / lookahead / context length.

3. The update-frequency sweep is helpful, but the explored settings are still limited, and the final conclusion remains fairly coarse (“moderately lazy is best”). The reviewer would like a clearer recipe or more principled guidance for selecting the update interval.

4. Scalability and deployment realism are not fully explored. The experiments provide limited insight into whether Aurora remains effective under more realistic multi-server deployments, bursty traffic, or more heterogeneous request streams.

5. The paper suggests that the main benefit comes from closing the online loop, while discarded-branch supervision often provides only modest additional improvement in standard settings. This weakens the case for the added complexity of some components.

6. The paper does not provide an open-source implementation, and it offers relatively limited implementation details (e.g., the batching granularity, speculative-tree packing scheme, and the exact handling of accepted/rejected labels during loss computation), which makes it harder for reviewers to fully understand and assess a contribution that is primarily engineering-oriented.

---

> ### Author Rebuttal · Authors · 2026-03-31
>
> We thank the reviewer for the thorough and constructive evaluation. We are glad the reviewer finds the problem important and practical and the framework fairly complete. We address each weakness below.
>
> ### W1. Why mismatch arises?
>
> The paper goes beyond demonstrating that online adaptation works; it provides a clear systems-level account of why offline speculative training under-delivers in practice. We argue that mismatch arises because a speculator is optimized under data and runtime conditions that differ from those encountered at deployment time: real serving utility depends on system-level execution effects, the verifier itself may drift over time, and live traffic can deviate from the offline training distribution. Aurora is designed precisely to close these gaps by learning continuously from live traces under the actual serving stack.
>
> ### W2: Tree Attention Overhead
>
> We agree that quantifying Tree Attention cost is important. Our Tree Attention mechanism (Figure 2\) constructs a custom block mask that encodes the causal structure of the speculative tree. This allows processing all accepted and rejected branches in a **single batched forward and backward pass**, rather than iterating over each branch individually. The key efficiency insight is that the speculative tree produces a sparse attention pattern, and `flex_attention` automatically skips masked blocks at the hardware level, making it substantially faster than dense attention over the full tree.
>
> We have quantified this overhead in profiling. Specifically, average training time increases from 68 ms to 90 ms when tree attention is enabled with an average branching factor of 40, while the number of training tokens increases from 555 to 675. Despite this added cost, training throughput remains approximately 7,500 tokens/s, still about 21× faster than the inference throughput of 354 tokens/s. Therefore, in our setting, the standalone overhead of tree attention is low and does not become the end-to-end bottleneck.
>
> ### W3: Update Frequency Guidance
>
> We expand on the analysis in Figure 5 by providing a practical guideline for choosing the update interval. Based on our sweep over update intervals ranging from 48 to 1600 requests, we make the following observations:
>
> Recommended interval: 10× the inference batch size. In our experiments, update intervals of 80 (purple) and 40 (green) achieved nearly identical recovery in acceptance length, while 80 maintained strong throughput. An interval of 40 may still be further optimized to have minimal impact on inference latency. Exploring this trade-off is an important direction for future engineering work, particularly through more asynchronous speculator updates. More broadly, this result is consistent with prior experience in asynchronous RL systems, where updating weights at every step can be made practical with sufficiently efficient system design.
>
> ### W4: Scalability and Deployment Realism
>
> Experiments now include two additional models beyond Qwen3-8B:
>
> - **MiniMax M2.1** (229B, MoE 256 experts, top-8 routing): 4×H200 GPUs, TP=4, FP8; training on 1×H200. Aurora raises acceptance length to 2.8 and throughput **1.57×**.
>
> - **Qwen3-Coder-Next** (80B total / 3B active parameters, MoE with 512 experts, hybrid GatedDeltaNet \+ Gated Attention architecture): Served on 4×H200 GPUs with TP=4 and expert parallelism in FP8. Training is performed on a different node using 1×H200. Aurora raises acceptance length beyond 3.0 and achieves up to **1.51× throughput** improvement.
>
> Results are available in this [figure](https://anonymous.4open.science/r/990509mm/aurora.pdf).
>
> **End-to-end throughput under varying batch sizes** (TPS = tokens/s):
>
> | BS | Config | Mean | P50 | P05 | P95 | Speedup | Acc Len
> | :- | :- | :- | :- | :- | :- | :- | :-
> | *MiniMax M2.1* 40k steps |  |  |  |  |  |  |
> | 1 | w/o spec | 135 | 136 | 131 | 137 |  |
> | 1 | w/ spec | 212 | 211 | 163 | 270 | **1.57×** | 2.62
> | 8 | w/o spec | 79 | 79 | 74 | 85 |  |
> | 8 | w/ spec | 107 | 105 | 80 | 137 | **1.36×** | 2.62
> |16 | w/o spec | 65 | 64 | 59 | 72 |  |
> |16 | w/ spec | 83 | 83 | 61 | 112 | **1.29×** | 2.62
> |32 | w/o spec | 54 | 53 | 47 | 67 |  |
> |32 | w/ spec | 67 | 65 | 44 | 101 | **1.25×** | 2.62
> | *Qwen3-Coder-Next* 80k steps |  |  |  |  |  |  |
> | 1 | w/o spec | 176 | 178 | 172 | 178 |  |
> | 1 | w/ spec | 266 | 265 | 209 | 321 | **1.51×** | 3.06
> | 8 | w/o spec | 120 | 122 | 105 | 135 |  |
> | 8 | w/ spec | 146 | 144 | 110 | 190 | **1.23×** | 3.07
> |16 | w/o spec | 100 | 102 | 75 | 119 |  |
> |16 | w/ spec | 108 | 104 | 76 | 157 | **1.09×** | 3.06
>
> MiniMax M2.1 uses lookahead 4; Qwen3-Coder-Next uses lookahead 5.
>
> These results demonstrate Aurora generalizes across dense models (Qwen3-8B), large MoE models (MiniMax M2.1), and hybrid MoE architectures (Qwen3-Coder-Next), with consistent speedups at small-to-moderate batch sizes.
>
> ### W5: Open-source Implementation
>
> We will release the full codebase shortly.

---

> > ### Author Rebuttal · Reviewer_Xu55 · 2026-04-03
> >
> > My concerns have been addressed. I hope to see the codebase open-sourced soon and will maintain my positive score.

---

### Official Review · Reviewer_yfJ7 · 2026-03-08

**Soundness:** 3
**Presentation:** 3
**Significance:** 3
**Originality:** 3
**Overall Recommendation:** 4
**Confidence:** 3

**Summary:**

The paper presents Aurora, a unified system that trains a speculative draft model online while it serves live traffic. It uses an asynchronous training server to update the draft model via GPU-to-GPU RPC, aiming to fix the mismatch between offline training and online serving.

**Compliance With Llm Reviewing Policy:**

Affirmed.

**Final Justification:**

The rebuttal addresses most of my questions well and I would love to raise my score.

**Key Questions For Authors:**

1. How does the separate speculator approach in Aurora remain competitive or preferable compared to self-drafting architectures (e.g., Medusa, EAGLE-3) that are naturally more compatible with prefix caching?

2. Can the authors provide a rigorous cost-benefit analysis showing that the runtime overhead of logging hidden states and logits for online training is superior to a high-frequency offline retraining loop using standard inference logs?

3. Given that SGLang already supports SpecForge for speculative adaptation, what are the specific algorithmic or system-level advantages of Aurora's RL-based loop that SpecForge does not provide?

4. How does the system maintain stability and performance under realistic, gradual, or overlapping distribution shifts, rather than the artificial, abrupt switches used in the Ordered Streams experiment?

**Limitations:**

Yes

**Strengths And Weaknesses:**

Strengths

+ Successfully implements an asynchronous, closed-loop pipeline for online distillation using TensorPipe and Hot-swap mechanisms. Resource Efficiency on Training Side: The Zero-Copy Target Model design allows the training server to update a ~1B speculator without loading the massive target model (8B-70B+), saving significant VRAM.


Weaknesses

- Weak Motivation vs. Self-Drafting Trends: The paper focuses on a separate speculator model. However, the industry is moving toward self-drafting (e.g., Medusa, EAGLE) where draft heads are part of the target model. Self-drafting is often better for prefix caching. The authors do not clearly explain why a separate speculator is still a better choice given these trends.

- Unclear Need for Online vs. Fast Offline: The system modifies the inference path to log and send hidden states and logits. This introduces runtime overhead in memory and bandwidth. There is no evidence that a fast offline loop (e.g., retraining every few minutes on logged data) wouldn't achieve the same results without the complexity of modifying the live inference engine.

- Lack of Comparison with SpecForge: SGLang already has SpecForge, which handles speculative model training/adaptation. The paper fails to distinguish Aurora's unique value from SpecForge or show why this RL-based approach is superior.

- Simple Distribution Shift Setup: The Ordered Streams (switching hard from one domain to another) is an artificial way to show shift. Real-world traffic is more mixed and subtle; the system's stability in a more realistic setting is not proven.

---

> ### Author Rebuttal · Authors · 2026-03-31
>
> We thank the reviewer and address each concern below.
> ## W1 & KeyQ.1: Self-drafting
> We agree that self-drafting is an important direction. Aurora is not limited to a standalone drafter; it is a unified online adaptation framework that supports multiple drafter forms. Aurora uses EAGLE-3 in all experiments. However, the EAGLE-3 drafter remains a trainable speculator: it consumes verifier hidden states, can be initialized from scratch while reusing the target model token embedding, and can be updated online. This distinction matters in practice because many frontier models, such as Kimi-K2.5 and GPT-OSS, do not have MTP heads. Even when such heads do exist, Aurora can still train its draft part online: continuous, traffic-specific adaptation.
> ## W2 & KeyQ.2: More runtime overhead
> We include the overhead of hidden-state transformation and weight synchronization. In this experiment, the speculator weights are kept unchanged, so the measured overhead captures only the runtime cost of transformation.
> System|TPS|P50 Latency (ms)|P99 Latency(ms)
> -|-|-|-
> Baseline (SGLang/EAGLE-3)|359|2840|3256
> Aurora (frozen)|355|2876|3303
> Aurora (trained)|500|2048|2409
> These results show that the logging overhead is small. At the same time, when the speculator is trained, it delivers substantial performance improvements.
> ## W3. Online vs Fast Offline
> Aurora introduces negligible inference overhead, whereas fast offline training requires either redundant target-model computation or substantial disk storage.
> EAGLE-3 training requires the target model intermediate hidden states. The current SOTA offline training system, SpecForge, provides two costly modes:
> - Mode 1: A separate data-preparation step runs the full target model forward pass, extracts hidden states, and saves them to disk. Each sample requires 64 MB of storage (48 MB aux hidden states + 16 MB final hidden states in BF16 at L=2048). These numbers increase for large models.
> - Mode 2: Avoids disk storage but loads the entire target model to compute hidden states and logits in real time, effectively doubling target-model GPU cost.
> It avoids both: hidden states are captured during inference verification and transferred GPU-to-GPU via asynchronous RPC, with no separate target-model forward pass or disk.
> ## W4  & KeyQ.3: Compare to SpecForge
> First, we clarify that Aurora builds on ideas made practical by SpecForge and complements it with online, closed-loop adaptation. Rather than competing with SpecForge, Aurora is designed to complement it by enabling online training and faster adaptation through insights from RL systems.
> ||Speculator Offline Training|Aurora (Online)
> |-|-|-
> |Architecture|Standalone|Unified training-serving system
> |Hidden state source|Mode1: Separate full target-model forward pass + disk storage target activations; Mode2: Full target model loaded on training GPUs|Byproduct of live inference verification, with minimal additional serving overhead
> |Target model on trainer|Mode1: loads LM head and runs it every batch; Mode2: loads full model (16 GB+)|Not needed— hidden states + logits, arrive via RPC from the inference server
> |Adaptation latency|Batch offline cycles|Continuous
> |Day0 serving|Requires pretrained draft model|Supported — train from random init on live traffic
> |Distribution shift|Must re-collect data and retrain|Automatic on-policy adaptation
>
> It adds four innovations beyond SpecForge.
> 1. On-policy learning: It trains on the speculator's own accept/reject outcomes under the live serving stack, capturing training-serving mismatch that offline distillation cannot observe.
> 2. Discard Sampling: It learns from accepted and rejected speculative branches. SpecForge trains only on target-model outputs. Rejections provide counterfactual feedback that teaches the drafter what not to propose, yielding a denser signal.
> 3. Closed-loop feedback: Its serve-to-train flywheel means the speculator continuously improves on exactly the traffic it serves. SpecForge trains on a fixed or periodically refreshed corpus, which drifts from serving conditions.
> 4. Minimal infrastructure cost: It requires no disk pipeline, no data versioning, no separate data-preparation jobs. The training loop runs as an asynchronous sidecar to inference engine.
> ## W5  & KeyQ.4: Distribution shift setup
> Our Ordered Streams setting is a controlled approximation of traffic patterns we observe in real serving workflows. In practice, online traffic is often non-stationary: one request type may become more prevalent while earlier patterns still remain, rather than switching abruptly. Because Aurora updates asynchronously, the speculator is naturally trained on a mixture of old and emerging traffic during transitions. We complement this with mixed-stream experiments in Sec. 4.1, where domains are continuously interleaved to heterogeneous live traffic. Together, these two settings capture two realistic deployment patterns—evolving traffic over time and mixed workloads—and Aurora remains stable in both.

---

> > ### Author Rebuttal · Reviewer_yfJ7 · 2026-04-04
> >
> > Thanks for the response and I would love the raise my score.

---

### Official Review · Reviewer_R4ZN · 2026-03-15

**Soundness:** 4
**Presentation:** 3
**Significance:** 4
**Originality:** 3
**Overall Recommendation:** 4
**Confidence:** 3

**Summary:**

This paper studies how to apply RL to continuously fine-tune the draft model in a speculative decoding setting. The results, based on a limited number of models, are promising, and the proposed design has been tested in SGLang, a commonly used LLM inference engine.

**Compliance With Llm Reviewing Policy:**

Affirmed.

**Key Questions For Authors:**

1. Can you show more pairs of verifier and draft models? I am particularly interested in larger MoE models. How does the adaptation cost grow with the draft model size and the verifier model size?

2. How does your approach behave as context lengths increase?

3. How does your approach behave in agentic and multimodal settings?

**Limitations:**

Yes

**Strengths And Weaknesses:**

Strengths:

1. This is a new and meaningful problem setup. Allowing the draft model to adapt online to dataset distribution shift and also to the capability of the verifier model is indeed important and practically relevant.

2. The RL-based fine-tuning idea also makes sense to me. The overall setup feels practical, and it is good that the paper includes ablation studies to justify the design choices.

3. The evaluation results are promising. It is nice to see clear benefits on Qwen-8B with a commonly used EAGLE-3 draft setup.

Weaknesses:

1. Only Qwen-8B is evaluated. Could you show more verifier–draft pairs? For example, GPT-OSS-120B with EAGLE-3, or a larger Qwen model such as Qwen-300B paired with Qwen-4B or some other draft model. It would be good to see how well the approach generalizes, and also how the adaptation cost scales across different verifier and draft configurations.

2. How does the adaptation cost scale with longer context lengths, for both input context and generated output length? Does the online-tuned draft model become more helpful in long-context settings, or not? I would really like to see results on this.

3. How does this approach behave in agentic settings? Do you have any results to support that case? Also, for the Qwen-8B setup, is this the text model or the VL model? It would be useful to understand how the method behaves in agentic and multimodal settings as well.

---

> ### Author Rebuttal · Authors · 2026-03-31
>
> We thank the reviewer for the encouraging assessment and for recognizing our work as a new and meaningful problem setup with excellent soundness and significance. Below we address weakness and question.
>
> ### W1 & Key Q.1: More Verifier-Draft Pairs
>
> We agree that broader model coverage strengthens the paper. We have added experiments on two recent frontier open-source models, in addition to the original Qwen3-8B results:
>
> - **MiniMax M2.1** (229B parameters, MoE with 256 experts, top-8 routing): Served on 4×H200 GPUs with TP=4 in FP8. Training is performed on a different node using 1×H200. Aurora (Scratch) increases acceptance length to 2.8 and delivers up to **1.57× throughput**.
>
> - **Qwen3-Coder-Next** (80B total / 3B active parameters, MoE with 512 experts, hybrid GatedDeltaNet \+ Gated Attention architecture): Served on 4×H200 GPUs with TP=4 and expert parallelism in FP8. Training is performed on a different node using 1×H200. Aurora raises acceptance length beyond 3.0 and achieves up to **1.51× throughput** improvement.
>
> Results are available in this [figure](https://anonymous.4open.science/r/990509mm/aurora.pdf).
>
> **End-to-end throughput under varying batch sizes** (TPS \= tokens-per-second):
>
> | BS | Config | Mean | P50 | P05 | P95 | Speedup | Acc Len |
> | :- | :- | :- | :- | :- | :- | :- | :- |
> | *MiniMax M2.1 (FP8)* online 40k steps |  |  |  |  |  |  |  |
> | 1 | w/o spec | 134 | 136 | 131 | 137 | – | – |
> | 1 | w/ spec | 211 | 211 | 163 | 270 | **1.57×** | 2.62 |
> | 8 | w/o spec | 79 | 79 | 74 | 85 | – | – |
> | 8 | w/ spec | 107 | 105 | 80 | 137 | **1.36×** | 2.62 |
> | 16 | w/o spec | 64 | 64 | 59 | 72 | – | – |
> | 16 | w/ spec | 83 | 83 | 61 | 112 | **1.29×** | 2.62 |
> | 32 | w/o spec | 53 | 53 | 47 | 67 | – | – |
> | 32 | w/ spec | 67 | 65 | 44 | 101 | **1.25×** | 2.62 |
> | *Qwen3-Coder-Next (FP8)* online 80k steps |  |  |  |  |  |  |  |
> | 1 | w/o spec | 176 | 178 | 172 | 178 | – | – |
> | 1 | w/ spec | 265 | 265 | 209 | 321 | **1.51×** | 3.06 |
> | 8 | w/o spec | 119 | 122 | 105 | 135 | – | – |
> | 8 | w/ spec | 146 | 144 | 110 | 190 | **1.23×** | 3.07 |
> | 16 | w/o spec | 99 | 102 | 75 | 119 | – | – |
> | 16 | w/ spec | 107 | 103 | 75 | 157 | **1.09×** | 3.06 |
>
> MiniMax M2.1 uses lookahead 4; Qwen3-Coder-Next uses lookahead 5.
>
> These results demonstrate Aurora generalizes across dense models (Qwen3-8B), large MoE models (MiniMax M2.1), and hybrid MoE architectures (Qwen3-Coder-Next), with consistent speedups at small-to-moderate batch sizes.
>
> ### W2 & Key Q.2: Context Length Scaling
>
> **a. Adaptation cost at longer generation lengths**
>
> We report the overhead of hidden-state/logits transformation and weight synchronization in our measurements. In this experiment, the speculator weights are kept unchanged (Aurora speculator frozen), so the measured overhead reflects only the runtime cost of transformation and synchronization, without any effect from weight updates.
>
> | Generation Length | System | Throughput (TPS) | Latency P50 (ms) | Latency P99 (ms)
> |:-|:-|-:|-:|-:
> | 1024 | Baseline (SGLang + EAGLE-3) | 359 | 2840 | 3256
> | 1024 | Aurora (speculator frozen) | 355 | 2876 | 3303
> | 1024 | Aurora (speculator trained) | 500 | 2048 | 2409
> | 2048 | Baseline (SGLang + EAGLE-3) | 370 | 4943 | 6257
> | 2048 | Aurora (speculator frozen) | 361 | 4985 | 6400
> | 2048 | Aurora (speculator trained) | 504 | 3662 | 4719
>
> As shown in the table, the hidden-state/logit transfer and weight sync overhead remains small as generation length increases.
>
> **b. Does the online-tuned draft model become more helpful in long-context settings or not?**
>
> Conceptually, Aurora should benefit more at longer contexts because: (1) each sample provides more training signal (more tokens per gradient step), (2) the offline storage cost scales linearly with context length — making offline pipelines even more impractical (64 MB/sample at 2048 tokens → 256 MB/sample at 8192 tokens), and (3) longer sequences exhibit more diverse token patterns, providing richer adaptation signal.
>
> We also report the convergence speed for different sequence lengths. Clearly, longer sequences clearly provide more training signals, leading to faster speedup.
>
> | Step | L = 1024 (TPS / Accept Length) | L = 2048  (TPS / Accept Length)
> | :- | :- | :- |
> | Baseline spec | 232/2.78 |  238/2.83
> | step200 | 244/2.96 | 246/3.02
> | step400 | 249/3.02 | 254/3.10
> | step800 | 250/3.06 |  255/3.11
> | step1200 | 257/3.14 | 264/ 3.20
> | step1600 | 254/3.13 |  260/3.19
> | step3200 | 261/3.19 | 266/3.23
>
> ### W3 & Key Q.3: Agentic settings
>
> We report the inference-speed dynamics of two frontier agentic models, MiniMax 2.1 and Qwen3-Coder-Next on agentic dataset using batch size = 8.
> | Step | MiniMax 2.1 (TPS / Accept Length) | Qwen3-next-80B  (TPS / Accept Length)
> | :- | :- | :-
> | No spec | 79/- |  119/-
> | step200 | 46/1.32 | 108/1.62
> | step400 | 64/1.50 | 114/1.84
> | step800 | 77/1.78 |  127/1.96
> | step1200 | 88/1.85 | 131/ 2.10
> | step1600 | 90/1.89 |  155/2.46
> | step3200 | 105/2.09 | 187/2.74

---

> > ### Author Rebuttal · Reviewer_R4ZN · 2026-04-05
> >
> > Thanks for the response. I remain positive about this paper, and please incorporate your newly added results in the later version of this paper. I believe they will largely strengthen the long-term soundness of your work.

---

### Decision · Program_Chairs · 2026-04-30

**Decision:**

Accept (regular)

**Comment:**

This paper addresses an important and practically relevant problem in speculative decoding: closing the training–serving loop so that the drafter can adapt online under shifting live traffic. The reviewers consistently viewed this as a meaningful contribution, and the paper’s unified training-serving design is a clear strength. In particular, the integration of asynchronous hot-swapping, online learning from accepted and rejected branches, and a practical serving-oriented implementation makes the work both novel and likely to be useful to the community.

The empirical results are also strong. The paper shows convincing gains for day-0 deployment, adaptation under mixed and shifting traffic, and improvement over static speculative decoding baselines. The rebuttal further strengthened the submission by addressing concerns about overhead and generality, including additional evidence that runtime overhead is small and that the method extends beyond the initial single-model setting.

The main weaknesses are limited theoretical depth and incomplete discussion of when continuous adaptation is most necessary in real deployments. While the paper motivates the adaptation–stability tradeoff well, the analysis remains more empirical and systems-driven than principled. It would also strengthen the final version to better clarify how often real-world serving traffic exhibits the degree of distribution shift that truly requires continuous drafter adaptation. In addition, because the contribution is heavily systems-oriented, releasing code would significantly improve transparency and impact.

Overall, the paper presents a compelling unified framework with clear practical value, strong experimental support, and a rebuttal that resolved most reviewer concerns.